# Motion magnification analysis of microscopy videos of biological cells

**Oren Shabi[1], Sari Natan[2], Avraham Kolel[3], Abhishek Mukherjee[4], Oren Tchaicheeyan[2], Haguy Wolfenson****[4], Nahum Kiryati****[1], Ayelet Lesman[2] ***

1 School of Electrical Engineering, Faculty of Engineering, Tel Aviv University, Tel Aviv, Israel, 2 School of Mechanical Engineering, Faculty of Engineering, Tel Aviv University, Tel Aviv, Israel, 3 Department of Biomedical Engineering, Faculty of Engineering, Tel Aviv University, Tel Aviv, Israel, 4 Rappaport Faculty of Medicine, Technion-IIT, Haifa, Israel

* ayeletlesman@tauex.tau.ac.il

## Abstract

It is well recognized that isolated cardiac muscle cells beat in a periodic manner. Recently, evidence indicates that other, non-muscle cells, also perform periodic motions that are either imperceptible under conventional lab microscope lens or practically not easily amenable for analysis of oscillation amplitude, frequency, phase of movement and its direction. Here, we create a real-time video analysis tool to visually magnify and explore sub-micron rhythmic movements performed by biological cells and the induced movements in their surroundings. Using this tool, we suggest that fibroblast cells perform small fluctuating movements with a dominant frequency that is dependent on their surrounding substrate and its stiffness.

## 1. Introduction

Periodic movements of cells have been observed in different biological situations. For example, isolated heart cells cyclically beat at a frequency of about 1 Hz, when cultured on Petri dishes or embedded in 3D hydrogels [1, 2]. Other types of cells, such as flagellated spermatozoa [3] and motile ciliary cells [4] were reported to beat at a higher frequency in the range of 1–40 Hz and 15–25 Hz, respectively. Moreover, it was suggested that non-cardiac cells may also perform active rhythmic contractile and dynamic shape changes when cultured on 2D substrates or embedded in 3D matrices. Examples include fibroblasts [5–9], neutrophils [10, 11], mast cells [12, 13], epithelial cells [14], neural stem cells [15]; all were shown to perform various kinds of oscillatory motions in the range of 0.001–0.1 Hz. Interestingly, even unicellular organisms, such as Saccharomyces cerevisiae and Dictyostelium discoideum exhibit shape dynamics [16]. For example, S. cerevisiae may show fluctuating nanometric movements (cell wall) in the range of 800–1600 Hz as measured by AFM [17], however, this movement is clearly not visible to current video microscopy.

On the organism level, one may mention as an example, the early phase of dorsal closure that is an important morphogenetic process during the embryonic development of Drosophila (fruit fly). There, apical cell-shape oscillations were observed in amnioserosa epithelial cells [18]. This pulsed—actomyosin-based—constriction frequency band lies between 2.4 and 9.0

**Funding:** This project was supported by the Zimin Institute for Engineering Solutions Advancing

Better Lives at Tel-Aviv University, Israel (http://tau.
ziminstitutes.org/), given to the last two authors
(N.K. and A.L). H.W. acknowledges support from
the Israel Science Foundation (1738/17; https://
www.isf.org.il/#/) and from the Rappaport Institute
(http://www.rappaport.org.il/Home). H.W. is an
incumbent of the David and Inez Myers Career
Advancement Chair in Life Sciences. The funders
had no role in study design, data collection and
analysis, decision to publish, or preparation of the
manuscript.

**Competing interests:** The authors have declared
that no competing interests exist.

mHz (periods of 110–500 s), and center at ~4.2 mHz fluctuations (~4 min period). The
mechanics of cell oscillation is driven by biochemical signaling through an intracellular nega-
tive feedback loop where the dynamics of the PAR proteins that interact with each other ulti-
mately regulate the apicomedial actomyosin assembly and disassembly with a period of ~4
min [19]. Laser microsurgery techniques were used to investigate observed dynamic oscilla-
tions in amnioserosa cells [20]. These oscillations are mostly out of phase in neighboring cells.
Both the contraction and expansion phases of this cycle are largely cell autonomous [20]. Inter-
estingly, actomyosin-based oscillations also have been observed in other biological systems at
low-Reynolds number. For example, shape oscillations were observed in purified preparations
of actin and myosin that exhibit spontaneous oscillations [21].

Cells show a variety of periodic shape changes, such as cortical shape oscillations and cell
membrane protrusion—retraction cycles. Specifically, cells generate sub-micron-size fluctua-
tions of focal adhesions (i.e., molecular contact sites between cells and their environment) that
allow them to sense the rigidity of their local surrounding environment [22, 23]. Indeed, even
suspended fibroblast cells between two optically trapped beads generate myosin dependent
force fluctuations in a wide bandwidth (0.1 to 10 Hz) that result in fluctuating cell shapes [24].
Also, shape oscillations of non-adhering (i.e., not attached to a substrate, free floating) fibro-
blast cells were observed to occur at a frequency of ~0.03 Hz [12, 25, 26].

The inner machinery of cells is the underlying engine that powers the observed periodic
movements and shape fluctuations. This includes the coupling between mechanochemical
cycles (actin polymerization waves), contractile actin-myosin cycles through periods of assem-
bly, activity and disassembly, and asynchronous activations of multiple motor protein clusters
causing unbalanced tractions [27–32].

Not only cells have the inner machinery to perform oscillatory movement, they are also sen-
sitive and responsive to rhythmic mechanical cues [33–40]. Even when shape changes are not
directly observed, one may detect sensitivity of circadian clock rhythms to extracellular matrix
stiffness, for example in the case of keratinocytes and epithelial cells [41].

On the technological side, fluorescent microscopy technology is advancing in terms of
lower photodamage, higher resolution and frame acquisition speed. Recently, for example, a
new microscopic tool was developed that allows high-resolution and high-speed volumetric
imaging, providing enhanced capabilities into real-time observation of cell dynamics [42, 43].
Our goal was to develop a video analysis tool to detect small (nano/sub-micron up to microm-
eter-scale) movements that are virtually unnoticeable under the microscope lens.

Video motion magnification methods have been developed in recent years [44] and have
found applications in variety of fields, from machine damage control (quality assurance of
mechanical systems) to patient monitoring (medical), as well as facial micro-expression recog-
nition (security). In this study, we aim to develop a new tool, based on the knowledge of motion
magnification, to the field of confocal optical microscopy. In this respect, Wu et el. [44] sug-
gested an Eulerian approach for movement magnification in video sequence that does not
require object or pattern detection, and thus avoids finding the motion before magnifying it.
Specifically, Eulerian video magnification (EVM) has been used to reveal temporal variations in
videos that are difficult or impossible to see with the naked eye, such as breathing chest move-
ment of an infant or flow of blood as it fills superficial arterioles on the face. Wadhwa et al. [45–
48] further improved the EVM method by making it faster, more accurate and less sensitive to
noise. Using Fourier decomposition, the motions were amplified in the phase domain while
avoiding amplification of the noise amplitude. Thus, EVM can also serve as a noise filter.

EVM amplifies movements only in a predefined frequency. In the case of biological cells,
the frequency of rhythmic motions is usually unknown beforehand and the cells may pulsate
in a wide bandwidth of frequencies. In addition, motion amplification of barely visible objects

can help in revealing object shape. For example, a video of two overlapping structural elements that slightly move at different frequencies. For this scenario, in some cases, time domain motion signals may not suffice as image elements may have almost identical movements. However, the difference could be significant in the frequency domain and therefore allow their distinction [49]. To answer this need, our EVM-based tool allows the user to automatically detect dominant movement frequencies in video microcopy image series and then allows to specify a chosen frequency value for motion magnification. A single frequency is chosen by the user based on a spectrum estimation method [50]. In addition, our tool allows the user to change the video processing parameters (*e.g.*, amplification factor or frequency to be amplified) in the graphical user interface (GUI), see S1 Fig. Our tool also contains in the GUI optional preprocessing steps for better visualization, such as image deconvolution due to the optical point spread function (PSF) of the image [51, 52]). In this study, we present the video processing methods, and discuss the validations and limitations of our tool. Finally, as an implementation example, we analyze the periodic movements of fibroblast cells cultured on glass dishes (2D culture) and cells fully embedded in a 3D hydrogel environment. As control, we analyze the movements of non-viable cells (chemically fixated). We detected dominant frequency in live cells that is significantly higher than non-viable cells. Furthermore, the standard deviation of the spectrum signal is typically higher for live cells, indicating more biological activity. We suggest that fibroblast cells on 2D glass substrate may perform slower periodic movements than cells embedded in a 3D gel environment. Finally, we analyzed movies of mda-mb-231 cells grown on top of a micropillar substrate, and noticed pillars periodically pulled by cells.

## 2. Methods

The developed tool combines several signal-processing methods to enable detection and estimation of small periodic movements in a video sequence collected by a microscope in biological laboratories. Specifically, we use confocal fluorescent microscopy to acquire volume-stack images of single fibroblast cells embedded in fibrin gels. This gel is commonly used to grow cells in 3D soft matrix, which reflects tissue environment [53, 54]. The process is schematically illustrated in Fig 1.

### 2.1 Biological sample preparation

**2.1.1 Cell culture.** Actin-GFP 3T3 fibroblast cells (a gift from Prof. Scott E. Fraser, USC, Los Angeles, CA) were cultured in DMEM supplemented with 10% fetal bovine serum, nonessential amino acids, sodium pyruvate, l-glutamine, 100 units/ml penicillin, 100 μg/ml streptomycin, and 100 μg/ml neomycin (all the materials supplied by Biological Industries, Kibbutz Beit Haemek, Israel), in a 37°C humid incubator.

**2.1.2 3D fibrin gel preparation.** Actin-GFP 3T3 fibroblast cells ($5 \times 10^3$ cells) were mixed with 10 μl of a 20 U/ml Thrombin solution (Omrix Biopharmaceuticals). Then, 10 μl of a 10 mg/ml fluorescently labeled fibrinogen (Omrix Biopharmaceuticals) suspension—labeled with Alexa Fluor 546 as we described previously [54]—was placed on a #1.5 coverslip in a 35-mm dish (MatTek Life Sciences) and mixed with the cells+Thrombin suspension. The resulting fibrin gel was placed in the incubator for 20 min to polymerize, after which, warm medium was added to cover the gel.

**2.1.3 2D rigid substrate sample preparation.** Actin-GFP 3T3 fibroblast cells were sparsely distributed on glass bottom dishes, culture treated, uncoated (14 mm glass diameter, #1.5 Coverslip, MatTek, Ashland, MA). We used paraformaldehyde (PFA)-fixated cells as a control group (at least 9 sample repetitions for each group). The cells adhered directly to the glass dishes.

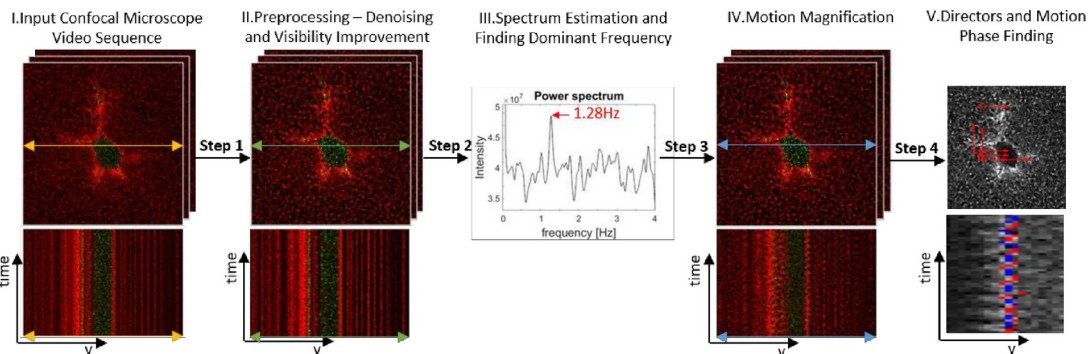

**Fig 1.** Overview of the method, including the steps: (I) Obtain an Input Video (time-lapse sequence) from a confocal microscope, choose a plane from the z-stack to analyze. A representative 'y-t plot' of pixel intensity versus time (aka, kymograph plot) is also shown below for the intersecting arrow line in the image above. The example shows confocal image of an isolated fibroblast cell (dark green pixels) embedded within Fibrin hydrogel 3D matrix (red); next, (II) Preprocessing of the input video for denoising, reversing the distortion of the microscope and video stabilization, a similar 'y-t plot' is also shown below for the same line as in the first step; next, (III) Finding dominant movement frequencies using spectrum estimation. The power spectrum in the example exhibits a detected dominant frequency at 1.28 Hz; next, (IV) motion magnification at a chosen dominant frequency (e.g., 1.28 Hz) using EVM, the kymograph below corresponds (as in 1st & 2nd steps) to pixel intensity change over time and now a wavy pattern is noticeable for some pixels; next (V) Detecting motion directors (in-plane) and motion phase (red/blue for a given director).

**2.1.4 2D cells on top flexible micropillars.** Live MDA-MB-231 epithelial cells (human breast adenocarcinoma source) were obtained from Prof. Yuval Shaked (Technion, Israel). These cells were cultured at 37°C in a 5% $CO_2$ incubator in DMEM supplemented with 10% fetal bovine serum, 100 IU/ml penicillin–streptomycin, 2 μM L-glutamine and 2 μM HEPES. One day before spreading experiments, cells were sparsely plated on a culture dish to minimize cell–cell interactions before re-plating on the pillars. The following day, cells were trypsinized using TrypLE (Biological Industries), centrifuged with growth medium, and then resuspended and pre-incubated in in HBSS buffer supplemented with 20 mM Hepes for 30 min before the experiment. Cells were then plated on top of the micropillars substrate, as previously reported by Wolfenson et al. [23, 55].

**2.1.5 Time-lapse microscopy.** *2.1.5.1 Images of 3T3 fibroblast cells.* Fibroblast cells were imaged with a Zeiss 880 (Axio Observer) confocal laser scanning microscope, equipped with a ×40, NA = 1.1 water immersion lens (Zeiss) and a 30-mW argon laser (wavelengths 488 and 514 nm). Throughout imaging, cells were maintained in a 37°C 5% $CO_2$ incubation chamber. Confocal z-stacks were acquired at time intervals as mentioned for each video.

*2.1.5.2 Images of MDA-MB-231 cells.* Time-lapse imaging of cells spreading on the pillars was performed as described previously [55]. Imaging was performed using an inverted microscope (Leica DMIRE2) at 37°C using a ×63, NA = 1.4, oil immersion objective. Bright-field images were recorded every 10 s with a Retiga EXi Fast 1394 charge-coupled device camera (QImaging). The microscope and camera were controlled by Micro-Manager software [56]. To minimize photodamage to the cells, a 600-nm long-pass filter was inserted into the illumination path.

## 2.2 Video analysis

The whole analysis was implemented in a script coded in MATLAB (version 9.5.0, R2018b; The MathWorks, Natick, MA) and is freely available (for non-commercial use only) from the corresponding author by request, both source code and executable file. Interaction and execution of the code is facilitated by a GUI, also developed in MATLAB.

**2.2.1 Choosing a region-of-interest (ROI).** Before starting the EVM analysis, the user may wish to concentrate on a specific ROI in the video, as there may be multiple cells in the whole field of view. For this purpose, the user has the option to choose in the first image of the video four points (XY-plane) that define a rectangular ROI. By extracting this rectangular ROI from each subsequent image in the video, a new subset sequence of ROI video is obtained for further analysis.

**2.2.2 Choosing the main spectral channels for analysis.** Our tool is expected to work well with any common video format collected by any microscope type since we first import the original video and convert it to Matlab format using the Bio-Formats toolbox [57]. Conventional videos contain either one (grayscale) or three color channels (RGB); confocal microscopy may contain multiple color channels, depending on the fluorescent labeling scheme and exciting laser bandwidth. The GUI allows choosing one or more channels for further analysis. For a video of mechanically coupled elements (e.g., green cells and their surrounding red fibrous matrix), analysis of information from two color channels may be used to corroborate parameter estimation (e.g., oscillation frequency, or edge detection) based only on information from one of the channels

**2.2.3 Preprocessing for visibility improvement.** The GUI enables performing preprocessing steps for better visualization of the video sequence, including the steps of image deconvolution, and global video stabilization. Details on these two steps follow.

The light emitted from a specific optic plane of the specimen passes through the optical system, generating a convolution of the real image with the PSF of the microscope. Each microscope has its specific combination of optical system and optical path that determine its unique PSF. Thus the input video may be somewhat blurred. This effect can be reversed by deconvolving the input video images with the PSF. The actual PSF may be either known beforehand by the user or may be blindly guessed and iteratively estimated using an algorithm, thus obtaining a deblurred image that better describes the specimen. Specifically, we have implemented in the GUI, the Matlab command for blind deconvolution as described by Pawley [52, 58] knowing that adding information about the specific imaging system will improve the data [59]. S2 Fig shows an example of an image of a cell before and after deconvolution, using the GUI.

Video stabilization is an important video enhancement option which aims at removing shaky motion from videos caused by a relative movement (rigid body movement) between the camera and the whole sample. This movement may occur due to immersed sample drift. Video stabilization was achieved by implementing Matlab commands relevant to "Point Feature Matching" method. Essentially, the built-in Matlab algorithm performs these six steps: (i) Read frames from a movie file; (ii) Collect salient points (edges) from each frame; (iii) Select correspondences between points; (iv) Estimating transform from noisy correspondences; v) Transform approximation and smoothing; and vi) Run on the full video.

Note that the steps described above are optional for data improvement but are not necessary, as the user may upload to the GUI an already preprocessed video.

**2.2.4 Edge detection.** For simplicity of use and interpretation we choose to focus on a limited subset of pixels for further analysis of spectral frequency and direction of movement. This subset comprises of pixels identified as 'edge pixels' of the cell, *i.e.*, points in the image at which the image brightness changes sharply or has discontinuities. The size of this subset is flexible and is amenable for control by the user (a knob in the GUI). Our motivation for choosing a limited subset of pixels for analysis is the notion that the cell performs its periodic motion mainly at the cell-edge by protrusive and retractive membrane movements (e.g., filopodial and lamellipodial). Therefore, we expect that the largest motions occur mainly along cell edges. To automatically detect the edge of the cell we first calculate the temporal mean (over a user defined time window) of grey levels for each pixel. This is done by finding the average intensity

for each specific pixel location in the video set of images. On the 'time averaged image', edge pixels are identified as pixels in which the spatial gradient is maximal. Eq 1 formulates the condition in which pixel (*x,y*) is considered an edge pixel if for every color channel *i*:

$$\left| \nabla \left( \frac{1}{N} \sum_{t=1}^{N} I_i[x, y, t] \right) \right| > GTH \qquad \text{(Eq 1)}$$

where $I_i$ is the intensity at (x, y) in a frame obtained from channel *i*; the number of frames (*N*) in a temporal window (this window may contain the whole video), and *GTH* determines what gradient value is high enough to be considered an edge. The threshold value is adjustable and set by the user, and its value is selected such that most relevant edges are detected. The user has the option to choose the method for edge-detection calculation out of six methods (available in Matlab 'edge' command): Sobel (default), Prewitt, Roberts, Laplacian of Gaussian, Zerocross, and Canny. For example, the Canny edge detector [60] is superior to Sobel's since it is less sensitive to noise, and more likely to detect true weak edges, however, it is slower. Example image of edge contour detected for a fibroblast cell image appears in S3 Fig.

**2.2.5 Spectrum estimation.**   To estimate the motion power spectrum, *i.e.*, the power spectrum of pixel intensity changes of edge pixels, we assume that every edge pixel intensity is a sample series of a random process. Using sample series $x_j[t]$ from edge pixel *j*, an autocorrelation series estimator is applied:

$$\hat{R}_j[\tau] = \frac{1}{N} \sum_{t=1}^{N} x_j[t] \cdot x_j[\tau - t] \qquad \text{(Eq 2)}$$

where *τ* could range from 1 to *N*. Given the autocorrelation estimator, a consistent estimator for the power spectrum is:

$$\hat{S}[\omega, L] = \frac{1}{M} \sum_j DFT(\{\hat{R}_j[\tau]\}_{\tau=1}^{L}) \qquad \text{(Eq 3)}$$

where *M* is the number of edge pixels and *L* is the length of the subseries of the pixel signal to be analyzed. Assuming that different sample series (different pixel intensities) are uncorrelated, the variance of the estimator decreases as *M* increases. In addition, it is possible that during the video the cell changes its motion frequency, and therefore, we provide the option to choose a length *L* for the subseries of the signal.

The dominant oscillatory motion frequency is identified as the frequency of the global maximum point on the power spectrum, excluding the 'direct current' (DC) peak.

**2.2.6 Motion magnification.**   In this step we implement the real time algorithm for phase-based motion magnification [46] using the original pseudocode [61] as a guide. The user is required to enter three inputs to this algorithm: (i) A value for the motion amplification factor; (ii) The frequency of oscillations to be amplified, and (iii) An input video: this video may be either the original video as obtained by the microscope or a pre-processed one (*i.e.*, deconvolved, stabilized, etc.). Finally, the output of this algorithm is the amplified-motion video.

**2.2.7 Automatic detection of significant motion of edge pixels (its location & direction).**   After amplifying the motion, our tool provides an option to draw arrows pointing along the motion direction of moving edges, on the video sequence. To avoid calculating the full optical flow field of the video sequence [62] we used a lower resolution but faster method, as described below.

First, instead of analyzing single pixel movement, we are searching for motion relative to a grid. A grid line *G* is defined as a one-pixel wide line segment of length *l* and the location of its

pixels satisfies one of the conditions described in Eq 4:

$$(x, y) \in G \ if \begin{cases} l \cdot m < x < l \cdot (m+1); \ y = l \cdot n \\ \qquad\qquad or \\ x = l \cdot m; \ l \cdot n < y < l \cdot (n+1) \end{cases} \qquad \text{(Eq 4)}$$

Where the length $l$ is characterized by an odd number of pixels that determines the grid density and $m, n \in \mathbb{N}$. A specific line segment $G$ contains motion if:

$$\sum_{t=1}^{N-1} \sum_{(x,y) \in G} |I[x, y, t] - I[x, y, (t+1)]| > MTH \qquad \text{(Eq 5)}$$

where $I$ is the intensity of pixels along $G$. The lower bound threshold $MTH$ in Eq 5 is determined by the user and is adjustable; its value is dependent on the noise level in the magnified video and is therefore chosen empirically.

Next, for every line $G$ that satisfies Eq 5, we aim to estimate the dominant direction of motion. For simplicity, we limited the automatic assignment of 'angle of motion' ($\theta$) to only four directions (45° apart, aligned relative to the image Y-axis, its vertical axis) and chose the one with the largest apparent motion amplitude. Respectively, we constructed four one-pixel wide line segments ($P_\theta$) of length $l$ pixels, centered on the geometric center (mid-length pixel) of the line segment $G$ and running along the $\theta$ direction. For each line segment $P_\theta$ we enumerate its pixels by ascending integer number $h$, where $-\frac{(l-1)}{2} \leq h \leq +\frac{(l-1)}{2}$ such that one end pixel gets the number $-\frac{(l-1)}{2}$ and the other end pixel gets the number $+\frac{(l-1)}{2}$. The motion amplitude is measured for each of the line segments $P_\theta(t)$. For each of the lines $P_\theta(t)$ the 'center-of-mass' point of the gray-level intensity in each time step is calculated, $P_{\theta CM}(t)$. The gray-level value (I[$\cdots$]) of the pixels along $h$ of the line $P_\theta$ represents virtual mass.

$$P_{\theta CM}(t) = \frac{\sum_h h \cdot I[P_\theta(t)]}{\sum_h I[P_\theta(t)]} \qquad \text{(Eq 6)}$$

The assigned 'direction of movement' $\theta_d$ is determined by:

$$\theta_d = \underset{\theta}{\operatorname{argmax}} \{rms[P_{\theta CM}(t) - \overline{P_{\theta CM}}]\} \qquad \text{(Eq 7)}$$

where $rms$ denotes the *root mean square* operation and $\overline{P_{\theta CM}}$ denotes the temporal average value of $P_{\theta CM}(t)$. Fig 2 presents the motion amplitude signal in four directions as a function of time for a simple case-study video—a pulsating circle changing its radius periodically.

**2.2.8 Finding movement phase.** After finding the dominant movement direction, we have provided a method to add indicators for the motion phase on a kymograph plot (*aka* 'y-t plot'); where the ordinate (*y*-axis) represents grayscale values for pixels in one chosen line of an image and the abscissa (*x*-axis) shows this same line data along progressing time. We have colored the center-of-mass pixel [$P_{\theta CM}(t)$; Eq 6] in either blue or red, depending on the sign of the temporal derivative of the center of mass signal, negative or positive, respectively (Fig 1 and S4 Fig. We estimate this derivative at each center-of-mass pixel by calculating the derivative of high-order (we used fifth order) fitted polynomial. This polynomial is an interpolation function of the center mass signal, $P_{\theta CM}(t)$, in an environment of $n$ neighboring center-of-mass pixels, on each side of the time series ($n$ prior and $n$ later in time). This number can be chosen by the user, and is influenced by the frame rate and the period of movement.

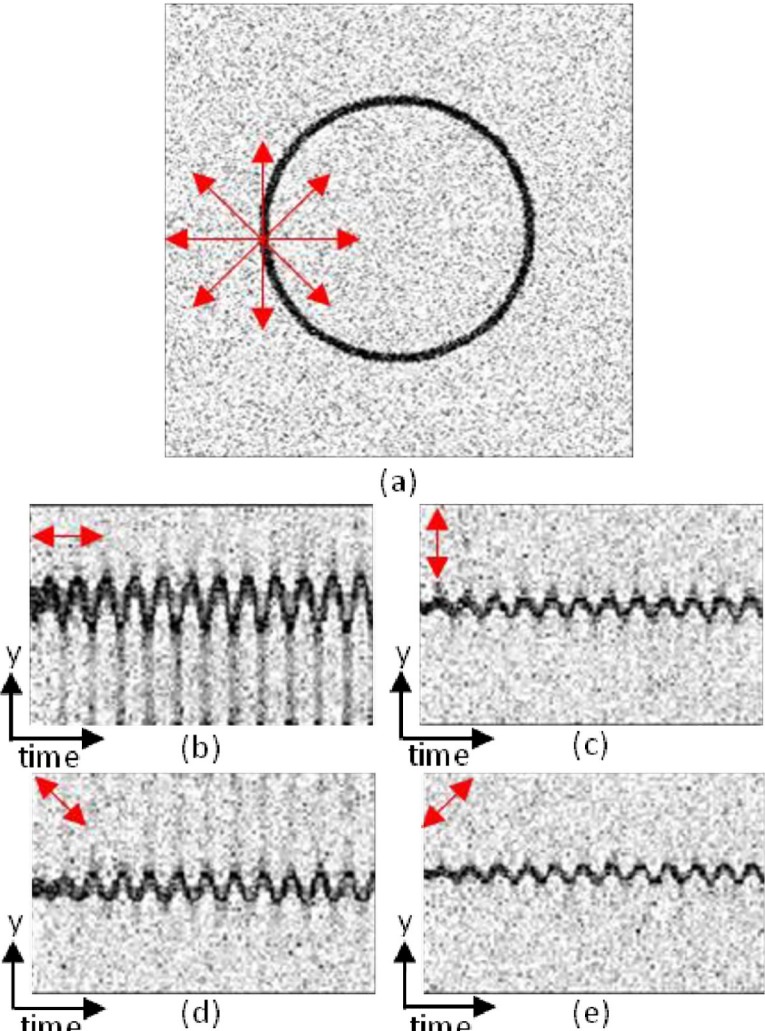

**Fig 2. Example video of a ring changing its radius harmonically.** (a) typical frame of the video and markers of the four analyzed directions, centered on an edge pixel that satisfies Eq 5. (b-e) kymograph plots along 0˚,90˚,135˚,45˚ direction, respectively.

## 3. Results

### 3.1 Validating example

To validate our tool, we used confocal microscope to acquire video sequence of a periodically stretched fibrin hydrogel. In order to stretch this soft gel under the microscope, we used a technique recently developed in our laboratory [63]. Briefly, the gel was polymerized within a cut-out silicone strip. The strip was then externally stretched by a motor, allowing to transfer controlled cyclic strains to the embedded gel. A fluorescently-labeled fibrin gel was stretched at a known frequency of 1Hz to several gel extension amplitudes in the range of 1 to 3 μm (implemented by 1˚ or 3˚ motion of the rotating step motor); all of the motion amplitudes are imperceptible in the raw data images by the naked human eye (for the 1μm stretch it was sub-pixel in size, where pixel size was 2.77 μm), see S1 Movie (Static gel) and S2 Movie (1˚ step stretch at 1 Hz). The ROI in these movies includes both gel and silicone rubber rim. Fig 3 presents the results of step 2.**2.5**. (Spectrum Estimation). The resulting spectra indicates that for a video

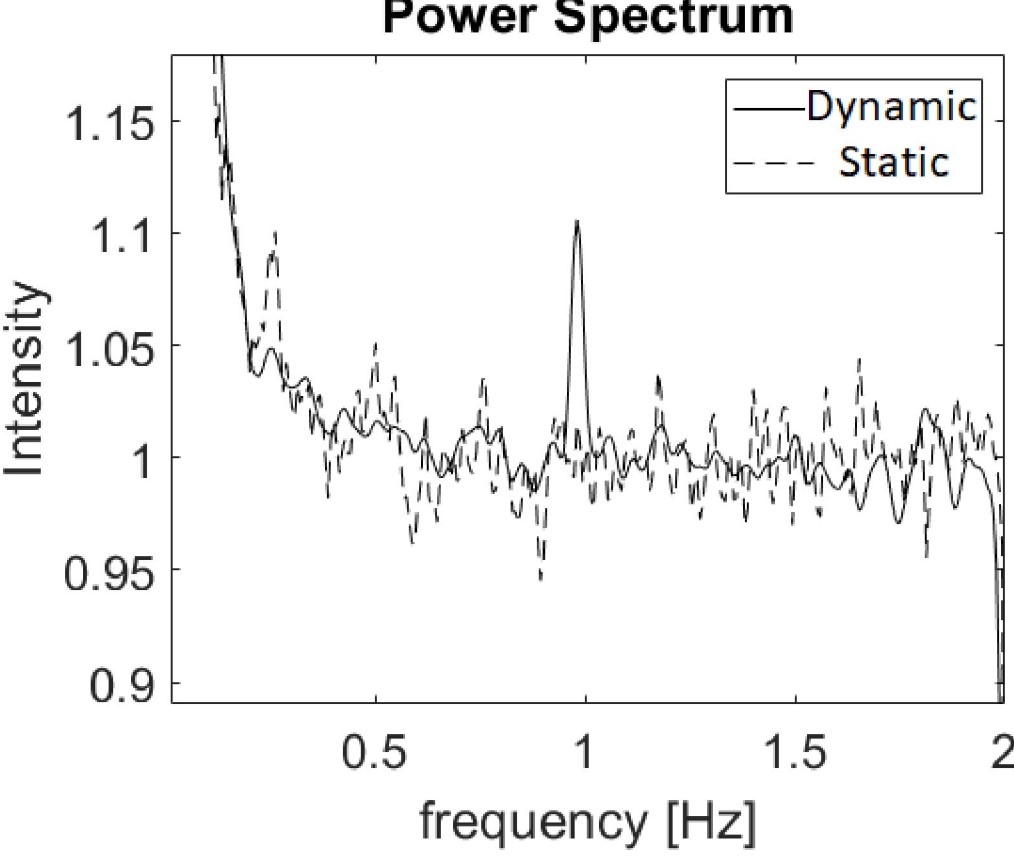

**Fig 3. Power spectrums of the motion in the gel stretching validation videos.** Static video (dashed line) and a dynamic spectrum video of a gel stretched by 1˚ step motor rotation–at 1 Hz (solid line).

with periodic gel movement the spectrum depicts a dominant spectral power peak at the correct frequency of 1 Hz and does not present any dominant frequency for a video of the same gel under static conditions (*i.e.*, non periodic stretch).

The pixel intensity noise component changes the gray-level of each pixel in a random manner. This noise component may have a wide temporal frequency bandwidth and therefore falsely represent periodic movements in that bandwidth that then will be amplified by the EVM algorithm. To validate that in the motion magnification step (section 2.2.6), the motion we amplify was real (*i.e.*, unique to periodically stretched samples) and not a result of noise, we estimated the amount of movement in a video sequence using the Lucas-Kanade optic flow algorithm [62]. Specifically, we used Matlab tools such as 'opticalFlowLK' and 'estimateFlow' with the threshold for noise reduction being 1E-4. The reported estimate for the motion was the total sum of the motion magnitude for all the pixels.

Fig 4 presents the result of motion as a function of the amplification factor for different stretching amplitudes. The result indicates that the motion magnification algorithm amplifies the noise, but to a lesser extent than the amplification of real movements. Due to noise amplification, a 'degree of motion' is demonstrated in a sample where the gel was held statically (*i.e.*, no periodic tension was exerted). For indication whether the amplified motions are real or not, one can estimate the 'degree of motion' (as quantified using the optic flow algorithm) in a sample of interest—for several values of the amplification factor—and compare the observed buildup of 'degree of motion' to the 'degree of motion' buildup calculated for a 'control'

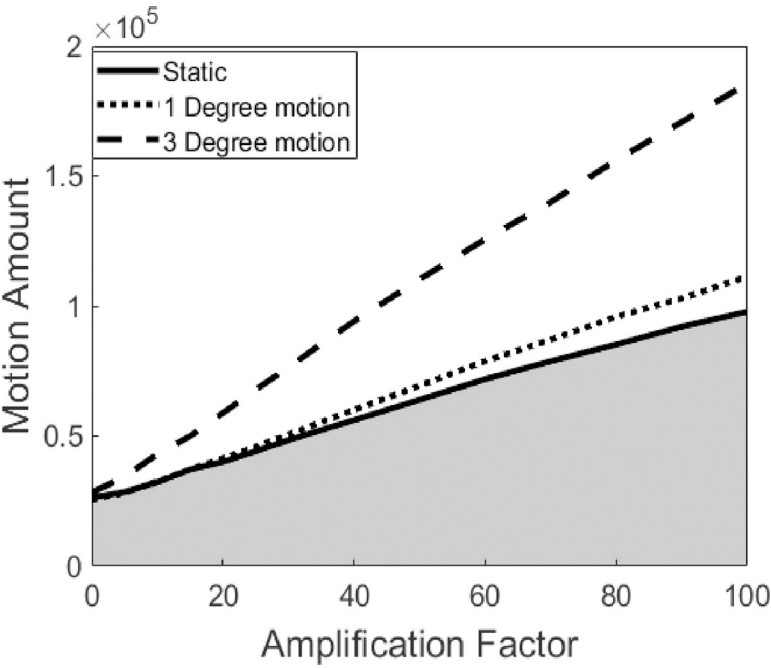

**Fig 4. 'Degree of motion' in amplified video sequence of a periodic stretched gel as function of amplification factor for three stretching amplitudes (0˚, 1˚ and 3˚ motion of the step motor).** The white region (the region above the continuous black line that limits the gray region) represents real motions and the gray region represents noise interpreted as movement.

sample that is essentially affected only by noise. Then one checks whether or not it falls within the "real" region on the motion as a function of amplification curve (white region in Fig 4).

## 3.2 Oscillatory movement behavior of fibroblasts cultured on top of 2D glass dishes vs. fibroblasts embedded in 3D hydrogels

Using the tool presented above we examined video sequences of fibroblast cells plated on 2D glass dishes and cells embedded in a 3D fibrin gel. We found that the frequency of the dominant motion of cells cultured on 2D dishes is about four times lower than for cells embedded in 3D soft fibrin hydrogel, with the average values: $\overline{f}_{dish} = 0.6 \pm 0.1\ Hz$ and $\overline{f}_{gel} = 2.3 \pm 1.3\ Hz$, respectively. The p-value for similarity between the groups is 0.02 which indicates significant difference. S5 and S6 Figs present the power spectra obtained for cells cultured on top of dish and for cells embedded in 3D soft fibrin hydrogel, respectively.

## 3.3 Oscillatory behavior of live vs. dead fibroblasts

Interestingly, due to image pixel's intensity noise, even samples containing dead cells (*i.e.*, PFA-chemically fixated) exhibit peaks (other than the dc-peak) in the power spectrum. Among these peaks the user may detect a 'dominant peak' that is higher than the spectral power of its close environment. We compared the significance of the dominant peaks in power spectra for groups of live and dead cells (S5 and S6 Figs) and have found that for live cells the value of the dominant frequency peak is 8% - 12% above the average peak height (8.2% in 2D; 9.2% in 3D) and the height of the dominant peaks found in videos of dead cells are only about 5%-8% above the average intensity (6.8% in 2D; 7.4% in 3D). This indicates that the movements in live cells are relatively higher than the noise induced apparent movements

in dead cells and that the detected dominant peaks in the power spectra for live cells are more significant (i.e., distinguished from their immediate neighborhood and higher in amplitude).

### 3.4 Micro-pillar oscillations by live mda-mb-231 cells

The micro-pillar setup, in which cells are cultured on pillar-array of defined length and stiffness, is a well-established platform to study cell-matrix interaction [64, 65]. Wolfenson et al. have shown that actomyosin-based contractile units activity between sub-micron pillars involves nanometer-scale rhythmic steps [23], and that long-term force application on micropillars is driven by local fluctuations in actin density [55]. As another demonstration of our tool's capability, we have analyzed videos of MDA-MB-231 cells grown on top of micropillars. Appearing in the Supplementary Movies: (i) an original video (S3 Movie) of MDA-MB-231 cells plated on top of micropillars; and (ii) an amplified video (Amplification factor = 20; S4 Movie). We have detected dominant peak frequency at about 0.009Hz, *i.e.*, a period of ~ 111 sec (S7 and S8 Figs). Looking at these movies one may notice that only micropillars underneath cells oscillate. The population of oscillating micropillars changes as the cell moves and extends. The direction of oscillation seems to correspond with cell alignment on the plane, and that waves of propagation along micropillars occur (*e.g.*, top of the image to its bottom and reverse). Finally, we have noticed that pillar oscillations may occur in bursts, *i.e.*, oscillation amplitude for a specific pillar is not constant throughout the video (time duration).

## 4. Discussion

We have developed an application for analyzing time-lapse image series produced by optical microscope, based on Eulerian approach to video motion magnification. To enable complete analysis of research data we incorporated into a GUI interface complimentary methods for spectral analysis and motion detection and its analysis, as well as, allowed for preprocessing step to prepare visually optimal data for the analysis. We demonstrate the use of our application in analysis of fibroblast cells cultured in various types of environments (dish, hydrogel), and of MDA-MB-231 cells cultured on top of micropillars.

### 4.1 Limitations

Our tool has some theoretical and practical limitations.

i) The input video frame rate and total duration determine together the frequency range and spectrum resolution. From Nyquist–Shannon sampling theorem, the highest frequency of the sampled signal is half the sampling rate and higher frequencies in the original signal will suffer from aliasing (aka, wagon-wheel) effect. In other words, if the video sampling rate is not high enough, we will interpret high motion frequencies as small ones. In addition, from the variance expression of Blackman-Tukey power spectrum estimator [50], as the signal duration is longer in time the estimator variance is getting smaller. Therefore, frequency estimation is improved as the signal is recorded for longer times. The longer the video acquisition period one may be able to detect lower frequency oscillations.

ii) Signal-to-noise level in confocal microscopes is dependent on multiple parameters [66] such as: pinhole diameter, type of detector, pixel size, illumination intensity, scan rate, and the image post-processing steps. Time lapse microscopy of living cells often requires fast acquisition of images and therefore relatively larger pixel size to be able to capture dynamic phenomena on the length scale of tens of microns. Noise in the video, *i.e.*, random changes in gray-level intensity over time, may be interpreted as motion in the EVM method, as noise contains a multitude of implicit frequencies. Therefore, with a high amplification factor we may erroneously synthesize motion from a video of a static object. To deal with it, for a given experiment

the user should examine the amount of "motion" in a static control measurement as function of amplification factor, allowing to determine its threshold (as we performed here, see Fig 4). Such 'static' control video may be of deliberately fixated cells or where any external movement and deformation is neutralized.

iii) One should maintain a watchful eye and notice the actual resulting amplified video to see if it indeed makes biological sense. Note that the EVM algorithm does not distinguish between gray level changes due to color level changes, for example in fluorescent protein expression levels *vs*. gray level changes due to actual movement in space.

iv) The user cannot always know beforehand what will be the dominant oscillation frequency that may change depending on several parameters. A fibroblast cell either embedded in 3D extracellular matrix or residing on top of a 2D substrate tests the rigidity of its immediate environment by applying forces on it. The 'rigidity sensing' mechanism was suggested recently to have a periodic nature (0.1 Hz to 0.001 Hz, range) [7, 67]. Note that the dominant frequency and oscillations bandwidth are sensitive to experimental setup, cell type and physiological state [67]. It is also known that the manner cells interact with rigid 2D substrates and with 3D soft environment may differ significantly.

v) Another limitation is the actual performance of the EVM algorithm used in our current implementation of this GUI tool. We used in this work a 1$^{st}$-order Taylor approximation based algorithm, where recently advanced algorithms allow for 2$^{nd}$- and 3$^{rd}$-order accuracy [68–70] that results in a better amplification of accelerated or jerked movements in the video, less blur in the constructed images, and neutralization of large and slow background movements [71].

## 4.2 Comparison to literature

The main motivation of this paper is to present a new applicative tool that combines several useful features for analyzing time-lapse microscopy of single living cells. Indeed, recently an EVM adaptation for live cell microscopy analysis of photo-conversion protocol time lapse images (HeLa cells) has been demonstrated [72]. However, the focus of that adaptation was lowering video image noise. In addition, their preprocessing step is different from ours, as they used a Laplacian pyramid for each image. Finally, the output of their application is noise-reduced image instead of motion amplified video with indication of movement region of interest, arrows of movement direction and indication of phase.

As a specific demonstration of the capabilities of our application, we show that cells directly attached to a Petri glass dish may slowly and periodically deform, relative to the same cells but embedded in a 3D fibrin matrix that may demonstrate faster oscillatory movements. Possible reasons are that glass dish (~GPa range) has a much larger stiffness than 3D hydrogel (~KPa), effective density of available sites for cell attachment to its substrate, and mode of attachment (through specific receptors or through nonspecific polar interactions). Engler et al. [73] had found that the beat rate of embryonic cardiomyocytes on stiff matrices (having scar-like stiffness) is lower, and eventually the cells stop beating after some time. Those cells beat best on a matrix with heart-like elasticity. This finding may corroborate our result for fibroblast cells, however, it is fully acknowledged that exploring this issue requires experimental work that is beyond the scope the current study.

We would remark that in 2D cultures the cells are attached to the glass bottom and the stiff substrate does not allow mechanical interaction between the relatively far away cells. In fibrin gels, cells embedded in the soft gel exert force which may be carried along relatively long distances in the elastic gel [74, 75], and therefore oscillations at the proximity of a cell may be caused by oscillations of the same cell and those exerted by neighboring cells. Hence, some

oscillatory movement may be mediated by neighboring cells and not necessarily by the cell itself. Furthermore, the relatively large mechanical compliance of the gel may allow the fast oscillatory action of the actomyosin machinery while the stiff glass bottom may halt the actomyosin machinery at isometric exertion (according to the force-velocity relationship, FVR [76]).

Another important feature of our tool is the ability to detect patterns of cell movements that are unidentifiable under normal imaging conditions. This is clearly demonstrated in videos of MDA-MB-231 cells plated on micropillar arrays. Such arrays are used for measuring cellular forces during the process of mechanosensing, which occurs when cells pull on the matrix (pillars) via integrin adhesions. In recent work, Wolfenson et al. [23, 55] found that this process occurs in a periodic fashion, typically at the cell edge. Interestingly, applying the EVM algorithm using our tool on videos of individual cells spreading on pillars shows two additional previously undetected features apart from the cell edge contractions. First, noticeable pillar contractions beneath the central area of the cell are observed, which, together with the cell edge contractions, indicate that significant mechanical power is invested in slow collective movements of all the pillars. Second, noticeable rounds of bursts of intense activity, followed by "quieter" round of activity, are observed, suggesting that a large-scale oscillatory mechanism might drive this process (S4 Movie). Our tool thus may provide a more complete view of the mechanosensing process, in ways that were previously unavailable.

Currently, tissue constructs with cells embedded in three-dimensional matrices that mimic the natural tissue serve as platforms for basic research, regenerative medicine and drug discovery. In such tissue constructs, force oscillations were detected, for example, as ensemble of beating cells movements [77]. Thus, the EVM tool may be used for the study of mechanobiology in such tissue constructs. It may also be used for the study of biochemical oscillations as exhibited, for example, by oscillatory enzyme kinetics in cyclic reactions [78].

## Conclusions

We have developed a reliable video analysis tool for research of periodic movements of living cells. In addition, we suggest a connection between cell motion frequency and its substrate properties. This tool may be used both for basic research and for diagnostics of pathological states.

## Supporting information

**S1 Movie. Fluorescent fibrin gel held statically.**
(WMV)

**S2 Movie. Fluorescent fibrin gel that is stretched dynamically (stretch-release cycles at 1Hz) at 1˚ rotation of a controlled motor.**
(WMV)

**S3 Movie. An original video of MDA-MB-231 cells plated on top of micropillars (each 2 μm in diameter).** Original frame rate is 1 frame per ten seconds. The movie is accelerated for visualization (fast-forward × 200 times).
(AVI)

**S4 Movie. EVM amplification of S3 Movie (Amplification factor = 20, frequency 0.009Hz).** Other details are the same as in S3 Movie.
(AVI)

**S1 Fig. Typical view of the GUI (Graphical User Interface).**
(DOCX)

**S2 Fig.** Typical frame of example video before (a) and after (b) deconvolution process.
(DOCX)

**S3 Fig.** Acquired image in a confocal microscope fibroblast cell (colored green pixels) embedded in fibrin gel (colored red pixels) (a); Edge detection: cell contour pixels detected (b); Spectrum Estimation: video power spectrum analysis for the edge pixels (c).
(DOCX)

**S4 Fig. Phase of movement.** Expansion designated by blue pixels, and contraction by red pixels, delineated on a kymograph of a short segment of a ring changing its size periodically.
(DOCX)

**S5 Fig. Representative power spectrum of fibroblast cells cultured on top of a 2D glass dish.** (A) Live cells with a dominant peak is at 0.76 Hz, and (B) dead cells (PFA fixated) without a distinguished dominant peak.
(DOCX)

**S6 Fig. Representative power spectrum of fibroblast cells embedded within a 3D fibrin gel.** (A) Live cells with dominant peak at 1.28 Hz, and (B) Dead cells (PFA fixated) without a distinguished dominant peak.
(DOCX)

**S7 Fig. Power spectrum of live MDA-MB-231 cells on top of micropillars.** The reference image for spectrum calculation is the whole image ($182 \times 286$ pixels).
(DOCX)

**S8 Fig. Power spectra of cells on micropillars.** The figure compares the power spectra of three windows ($70 \times 90$ pixels wide) that focus on micropillars: (i) 1st window, micropillars under a live cell (yellow); (ii) 2nd window, far-field micropillars without cells (red), taken from the same video of the live cells (yellow). (iii) 3rd window, micropillars under a fixated cell (blue), taken from the 'dead cells' video. Note that the blue and red spectra are very similar, especially in the high frequencies and both are very different (weaker) than the yellow spectrum of micropillars under live cells. Here, the reference image for spectrum analysis is smaller ($70 \times 90$ pixels) than the one shown on S7 Fig, in order to compare live MDA-MB-231 cells and far-field micropillars without any cells, in the same video.
(DOCX)

## Author Contributions

**Conceptualization:** Nahum Kiryati, Ayelet Lesman.

**Data curation:** Sari Natan, Avraham Kolel, Abhishek Mukherjee.

**Formal analysis:** Oren Shabi.

**Funding acquisition:** Nahum Kiryati, Ayelet Lesman.

**Supervision:** Nahum Kiryati, Ayelet Lesman.

**Validation:** Haguy Wolfenson.

**Visualization:** Oren Shabi.

**Writing – original draft:** Oren Tchaicheeyan.

**Writing – review & editing:** Haguy Wolfenson, Nahum Kiryati, Ayelet Lesman.

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
