## [Decision Letter · Decision Letter 0]

23 Jul 2020

PONE-D-20-13055

Motion Magnification Analysis of Microscopy Videos of Biological Cells

PLOS ONE

Dear Dr. Lesman,

Thank you for submitting your manuscript to PLOS ONE. After careful consideration, we feel that it has merit but does not fully meet PLOS ONE’s publication criteria as it currently stands. Therefore, we invite you to submit a revised version of the manuscript that addresses the points raised during the review process.

We look forward to receiving your revised manuscript.

Kind regards,

Jorge Bernardino de la Serna, Ph.D.

Academic Editor

PLOS ONE

Journal Requirements:

Reviewers' comments:

Reviewer's Responses to Questions

**Comments to the Author**

1. Is the manuscript technically sound, and do the data support the conclusions?

Reviewer #1: Yes

Reviewer #2: Yes

2. Has the statistical analysis been performed appropriately and rigorously? 

Reviewer #1: Yes

Reviewer #2: Yes

3. Have the authors made all data underlying the findings in their manuscript fully available?

Reviewer #1: Yes

Reviewer #2: Yes

4. Is the manuscript presented in an intelligible fashion and written in standard English?

Reviewer #1: Yes

Reviewer #2: Yes

5. Review Comments to the Author

Reviewer #1: The manuscript titled “Motion Magnification Analysis of Microscopy Videos of Biological Cells” has reported a video processing method to detect micromovement of cells. The authors demonstrated the utility of the method by analyzing periodic movements of fibroblast cells in 2D and 3D culture environments, and micropillar interactions with MDA-MB-231 cells. Their conclusions are clear and are appropriately supported by the data. Therefore, I recommended the manuscript for publication after minor revisions shown as below.

1. In figure 1, the steps description(I-IV) in the figure legend is not consistent with that in the figure. Please do the alignment.

2. Experimental details about cell culture in fibrin gel, culture dishes and micropillars need to be provided in method.

3. Although the authors performed comparison of oscillatory behavior between live and dead fibroblasts, I cannot find the spectra data for dead cells. Are they compared in 2D or 3D culture environment? The author also claimed that the motion frequency in 2D dishes is four times lower than in 3D hydrogel. Is it because of cell movement or the hydrogel movement? To clarify this, control experiments (fixed cells in 2D and 3D culture) should both be provided. In addition, fixed MDA-MB-231 cells in micropillar movements should also be tested as a control.

4. In figure S7, is it a spectrum for fibroblasts or MDA-MB-231 cells?

5. Movie S1 is not accessible

6. Movie S3 and S4 are not necessary, better to provide figures for clarification.

Reviewer #2: The authors present a very compelling analysis quantifying the oscillations that certain cells evert on their environments. The analysis tool is excellent, well explained, and well-validated, and the results are compelling and interesting.

This is the best quantification of this phenomenon of which the reviewer is aware, and the reviewer believes this to be an outstanding contribution that can be published in its current form.

If the authors choose to revise their discussion, it might make sense to connect with discussions of these oscillations in some additional contexts.

One is the well-known oscillations of the cells in a range of developmental contexts, beginning very early in development. A few nice references follow:

Sokolow, A., Toyama, Y., Kiehart, D.P. and Edwards, G.S., 2012. Cell ingression and apical shape oscillations during dorsal closure in Drosophila. Biophysical journal, 102(5), pp.969-979.

Hutson, M.S., Brodland, G.W., Ma, X., Lynch, H.E., Jayasinghe, A.K. and Veldhuis, J., 2014. Measuring and modeling morphogenetic stress in developing embryos. In Mechanics of Biological Systems and Materials, Volume 4 (pp. 107-115). Springer, Cham.

Durney, C.H., Harris, T.J. and Feng, J.J., 2018. Dynamics of PAR proteins explain the oscillation and ratcheting mechanisms in dorsal closure. Biophysical journal, 115(11), pp.2230-2241.

Another possible context is the early work from the group of Elliot Elson on using tissue constructs as platforms for basic research and drug discovery-- these oscillations are discussed as appearing in ensemble measurements on 3D tissue constructs. The current study is an outstanding and long sought quantification of these oscillations at the cellular level, and it might be nice to add the long history of efforts to see them to emphasize the importance and impact of the work.

However, the article as it stands covers much important ground, and these two angles are merely suggestions should the authors choose to incorporate them.

Great paper!

6. PLOS authors have the option to publish the peer review history of their article (what does this mean?). If published, this will include your full peer review and any attached files.

Reviewer #1: No

Reviewer #2: **Yes: **Guy Genin

---

## [Author Response · Author response to Decision Letter 0]

8 Sep 2020

Please see attached Response letter at the end of this PDF, thank-you.

---

## [Decision Letter · Decision Letter 1]

21 Sep 2020

Motion Magnification Analysis of Microscopy Videos of Biological Cells

PONE-D-20-13055R1

Dear Dr. Lesman,

We’re pleased to inform you that your manuscript has been judged scientifically suitable for publication and will be formally accepted for publication once it meets all outstanding technical requirements.

Kind regards,

Jorge Bernardino de la Serna, Ph.D.

Academic Editor

PLOS ONE

Additional Editor Comments (optional):

Reviewers' comments:

Reviewer's Responses to Questions

**Comments to the Author**

1. If the authors have adequately addressed your comments raised in a previous round of review and you feel that this manuscript is now acceptable for publication, you may indicate that here to bypass the “Comments to the Author” section, enter your conflict of interest statement in the “Confidential to Editor” section, and submit your "Accept" recommendation.

Reviewer #1: All comments have been addressed

Reviewer #2: All comments have been addressed

2. Is the manuscript technically sound, and do the data support the conclusions?

Reviewer #1: Yes

Reviewer #2: Yes

3. Has the statistical analysis been performed appropriately and rigorously? 

Reviewer #1: Yes

Reviewer #2: Yes

4. Have the authors made all data underlying the findings in their manuscript fully available?

Reviewer #1: Yes

Reviewer #2: Yes

5. Is the manuscript presented in an intelligible fashion and written in standard English?

Reviewer #1: Yes

Reviewer #2: Yes

6. Review Comments to the Author

Reviewer #1: The authors have addressed all comments in the revised manuscript. The only problem is that I can not find the supplementary figures in the manuscript may because of the uploading issue. As long as the authors put the figures, I would recommend the manuscript for publication. Thank you!

Reviewer #2: This is a great contribution. I loved it the first time, and it is even better now. Congratulations!

7. PLOS authors have the option to publish the peer review history of their article (what does this mean?). If published, this will include your full peer review and any attached files.

Reviewer #1: No

Reviewer #2: **Yes: **Guy Genin

---

## [Editor Report · Acceptance letter]

27 Oct 2020

PONE-D-20-13055R1 

Motion Magnification Analysis of Microscopy Videos of Biological Cells 

Dear Dr. Lesman:

I'm pleased to inform you that your manuscript has been deemed suitable for publication in PLOS ONE. Congratulations! Your manuscript is now with our production department. 

Kind regards, 

on behalf of

Dr. Jorge Bernardino de la Serna 

Academic Editor

PLOS ONE